# Recurrent neural networks as neuro-computational models of human speech recognition

Christian Brodbeck[1,2]*, Thomas Hannagan[2,3‡], James S. Magnuson[2,4,5]

**1** Department of Computing and Software, McMaster University, Hamilton, Ontario, Canada,
**2** Department of Psychological Sciences, University of Connecticut, Storrs, Connecticut, United States of America, **3** Stellantis Group, Poissy, France, **4** Computational Neuroscience, Basque Center on Cognition, Brain and Language, Donostia - San Sebastián, Spain, **5** Ikerbasque: Basque Foundation for Science, Bilbao, Spain

‡ Work conducted while the author was at the University of Connecticut.
* brodbecc@mcmaster.ca

## Abstract

Human speech recognition transforms a continuous acoustic signal into categorical linguistic units, by aggregating information that is distributed in time. It has been suggested that this kind of information processing may be understood through the computations of a Recurrent Neural Network (RNN) that receives input frame by frame, linearly in time, but builds an incremental representation of this input through a continually evolving internal state. While RNNs can simulate several key *behavioral* observations about human speech and language processing, it is unknown whether RNNs also develop computational dynamics that resemble human *neural speech processing*. Here we show that the internal dynamics of long short-term memory (LSTM) RNNs, trained to recognize speech from auditory spectrograms, predict human neural population responses to the same stimuli, beyond predictions from auditory features. Variations in the RNN architecture motivated by cognitive principles further improved this predictive power. Specifically, modifications that allow more human-like phonetic competition also led to more human-like temporal dynamics. Overall, our results suggest that RNNs provide plausible computational models of the cortical processes supporting human speech recognition.

## Author summary

Human brains deal with a constantly evolving stream of information acquired by the senses. The transitory nature of this input requires strategies for integrating information over time. Recurrent neural networks provide a computational model for this: Such networks maintain an internal state, and at each moment in time incorporate the new input into that internal state. Previous work has shown that

**Data availability statement:** The MEG data are contained in a public repository (https://openneuro.org/datasets/ds004276). Analysis code is hosted on GitHub repositories (https://github.com/christianbrodbeck/Eelbrain, https://github.com/christianbrodbeck/TRF-Tools, https://github.com/comp-cogneuro-lang/EARSHOT-MEG).

**Funding:** This work was supported by National Science Foundation grants BCS-2043903 and IIS-2207770 to JSM and CB. JSM's effort was also supported in part by the Basque Government through the BERC 2022-2025 program and by the Spanish State Research Agency through BCBL Severo Ochoa excellence accreditation CEX2020-001010-S and through project PID2020-119131GB-I00 (BLIS). The funders had no role in study design, data collection and analysis, decision to publish, or preparation of the manuscript.

**Competing interests:** The authors have declared that no competing interests exist.

such a model can simulate how humans recognize spoken words from a continuous stream of auditory input, developing general word recognition patterns that match findings from psycholinguistic experiments with human participants. While computational models of human word recognition typically used simplified inputs rather than real speech to reduce computational complexity, these new models can process real, acoustic speech signals. This allows us to compare the models' responses to speech with human brain responses to the same acoustic signals. We show that the temporal dynamics of the internal states of such networks resemble human brain activity. Furthermore, this resemblance increases when networks are designed in ways that are informed by psycholinguistic theory. Overall, our findings suggest that such recurrent neural networks can serve as models of the computational demands that humans face when listening to spoken language.

## Introduction

The computations underlying early auditory representations can be approximated as convolutional filters. Such mechanisms are typically studied using spectro-temporal receptive fields (STRFs) [1]. A majority of midbrain and early cortical neurons can be well described with such models, especially when incorporating some specific nonlinearities [2–5]. However, such mechanisms seem inadequate for capturing linguistic representations such as words. For example, STRFs have fixed temporal integration windows. In contrast, there is no hard upper limit to how long words can be, and words can be pronounced and recognized at drastically different speech rates.

Recurrent neural networks (RNNs) successfully solve such pattern detection problems in sequences that require integrating information over time [6], and may thus provide promising models for the neural mechanisms for human speech processing [7,8]. RNNs are highly incremental: they fully integrate the current input into their internal state at each processing step. This may make them particularly suited for modeling human speech perception, which is highly incremental [9–11]. Finally, long-short-term memory (LSTM), a particularly successful RNN architecture [12], may resemble the architecture of cortical microcircuits [13].

RNNs trained on phoneme symbol sequence inputs can simulate human sensitivity to phonotactic regularities in speech [14,15]. RNNs developed for automatic speech recognition can process real speech acoustic inputs [16–18]. However, these RNNs are not adequate models of human speech perception for several reasons: (1) these models typically process speech forward and backwards (with large portions of a signal simultaneously available in an image-like fashion), whereas humans are presumably limited to forward processing by the natural progression of time; (2) the RNNs' task is typically to transcribe a spoken input to a grapheme sequence, which is subsequently combined with an extraneous language model for transcription to words, whereas humans are thought to recognize phonological forms and words directly from speech, with simultaneous constraint from linguistic knowledge.

Recently, the RNN architecture has been adapted as a more realistic model of *human* speech recognition ("EAR-SHOT", [19]). An RNN was trained to recognize words, represented in the output as semantic feature vectors, directly from acoustic spectrograms. A simple architecture, consisting of a single hidden layer of LSTM nodes, was deliberately chosen to improve the potential for explainability. This model exhibited several characteristics of human speech recognition, such as the fine-grained time course of competition between words with overlapping phonology [9], and its hidden units exhibited phonetically organized responses to speech similar to those observed in human cortex [20,21], despite not being trained on phonetic targets.

Here we ask whether the temporal dynamics of computations in such an RNN also come to resemble the time course of human neural responses to speech. We address this by developing RNNs that can recognize words from the same acoustic stimulus sequences heard by participants in a magnetoencephalography (MEG) experiment, and predicting MEG responses from the RNNs' hidden unit activity (Fig 1). We compare the effects of different architectural decisions on the predictive power for brain activity to develop an RNN that most resembles human speech recognition.

In recent years it has been reported that the most proficient language models are also the best predictors of human brain activations [22]. Although Transformer models [23] currently dominate other architectures across virtually all language tasks, including speech transcription [24–26], this does not necessarily imply that they are inherently more "brain-like" [27]. Their dominance may instead reflect their computational efficiency on modern silicon hardware, which enables training of larger, deeper models on vast datasets—an advantage that may not correspond to neurobiological plausibility.

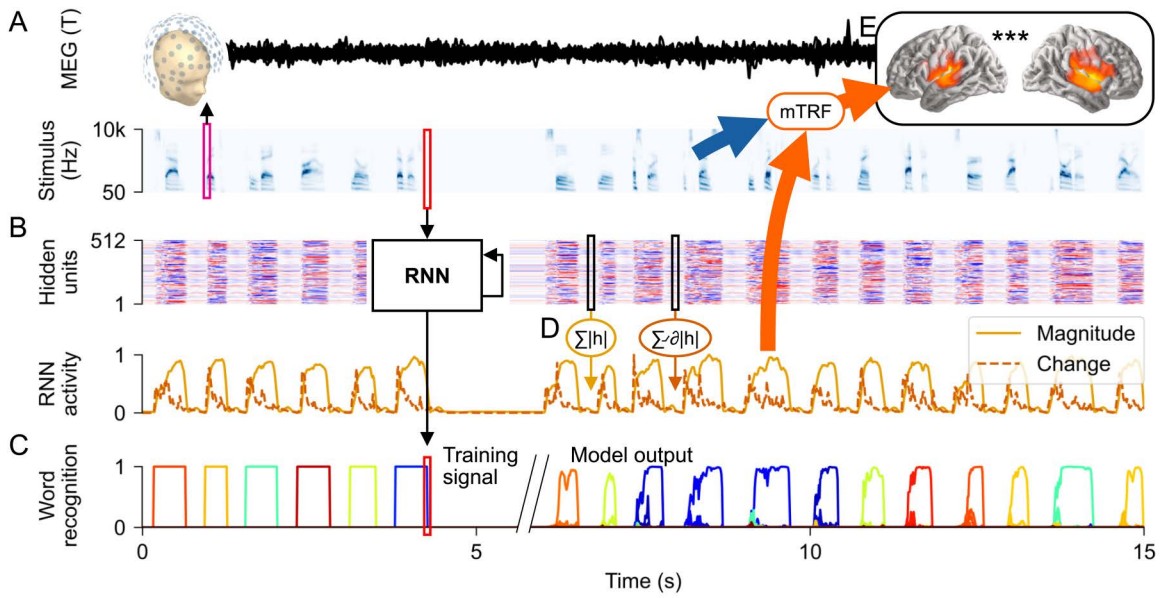

**Fig 1. General design: predicting brain activity from a recurrent neural network (RNN).** (A) Human participants listened to spoken words, while magnetic fields were measured with magnetoencephalography (MEG). (B) A recurrent neural network (RNN), trained to recognize words in arbitrary sequences of words and silence, processed the same stimulus sequence that each human participant heard. (C) The RNN was trained to output the word that is currently being heard, for the whole duration of the word ("Training signal"; in the localist output shown, each color represents a different word in the output space; all outputs are set to 0, except for the word that is currently in the input, i.e., each colored box represents the output corresponding to one word being set to 1 for the duration of that word). In practice, this is impossible in the early time course, because information about word identity is distributed over time in the acoustic input. Instead, the RNN tends to activate several possible candidates before settling on the right word ("Model output"; words are sorted phonetically, thus words with similar color have a similar onset). (D) RNN activity over time was quantified as the sum of hidden unit magnitude and the sum of hidden unit magnitude increases. (E) These two signals were then used to predict the source-localized MEG responses from each participant, while controlling for the predictive power of acoustic features (a gammatone spectrogram, an acoustic onset spectrogram, and word onsets). Brain responses were predicted through multivariate temporal response function (mTRF) models.

To our knowledge, there has been no systematic comparative study of Recurrent and Transformer architectures as predictors of human speech activations that controls for the number of floating point operations during training. The closest to this is Schrimpf et al. ([22], S9 Fig), who analyzed the impact on behavioral matching scores of several model variables related to the number of floating point operations during training (number of hidden layers and features, vocabulary size, training data size) for an extensive range of models, including 1 LSTM and 38 Transformers.

Furthermore, the attention mechanism, which makes Transformers so powerful for speech recognition, inherently entails the ability to compare information from all time steps in the input in parallel. Transformers thus circumvent a critical challenge that the human brain must solve: they process sound sequences as chunks of signal within which all the information is available at once, effectively converting the temporal dimension to a spatial one. In contrast, the auditory nerve provides auditory input to the brain strictly in the order in which it occurs in time. This is critical for simulating the temporal asymmetry in human word recognition: human listeners incrementally interpret speech in time. They need to store earlier input efficiently [11], and they may have to revise an earlier interpretation based on later input (so-called "right-context effects"). In contrast, a model that freely combines information across time can take the left and right context into account simultaneously, and without the need to store information for later access.

Here, our aim is to test an alternative to previous work, which predicted brain activity from state-of-the-art artificial intelligence engineering models [27,28]. Instead, we purposefully use artificial intelligence tools to design simple model architectures with the aim of relating them to human cognition in explainable ways.

## Results

To simulate human speech recognition, we trained LSTMs to recognize words from a gammatone spectrogram, a representation of sound that mimics the human peripheral auditory system. The input was a spectrogram, constructed by concatenating words and periods of silence. The trained output signal was a zero vector during silence, and a word-specific vector (differing depending on the output space, see below) for the entire duration of each word. Models were trained on long sequences of words and silence without resetting LSTM states. All models were trained to recognize 2934 words, selected to approximately include the lexical neighborhoods of the 1000 words which served as stimuli in the MEG experiment. For model training, each word was spoken by 15 synthetic talkers in addition to the human talker used for the MEG experiment. For each talker, a different 1/16th of words were withheld during training to serve as targets for evaluating model performance (i.e., the 1/16th withheld during training with a specific talker were not withheld with the other 15 talkers; note that it is not possible to test the model on completely novel words because the mapping from speech to semantics is mainly arbitrary).

### Sparse output spaces improve word recognition

What is the format of the target representation in human word recognition? One conjecture is that word recognition directly maps acoustic patterns to a semantic space [29]. Furthermore, lexical co-occurrence based vector spaces have been described as candidates for psychologically realistic representations of lexical semantics [30,31]. A model implementing this architecture is shown in Fig 2A: a dense layer projects the RNN output into the GloVe vector space [32], and the network is trained using the mean squared error (MSE) loss function. Alternatively, sound may be linked to an abstract intermediate level of *word forms* before being linked to semantics [33]. This was implemented using a localist (one-hot) output space, in which the output vector has the size of the lexicon, and each word corresponds to one of the elements of this vector. As is common for classification problems, the RNN was connected to the output through a dense layer with a sigmoid activation function, and was trained to minimize binary cross-entropy (Fig 2B). An additional possibility is that the meaning of each word corresponds to a sparse set of features; this was implemented using Sparse Random Vector (SRV) output [19], similar to the localist output except that each word corresponded not to a single vector element, but to 10 randomly selected elements of a vector of length 300 or 900 (the original model used 10 out of 300 for a 1000 word lexicon,

PLOS Computational Biology

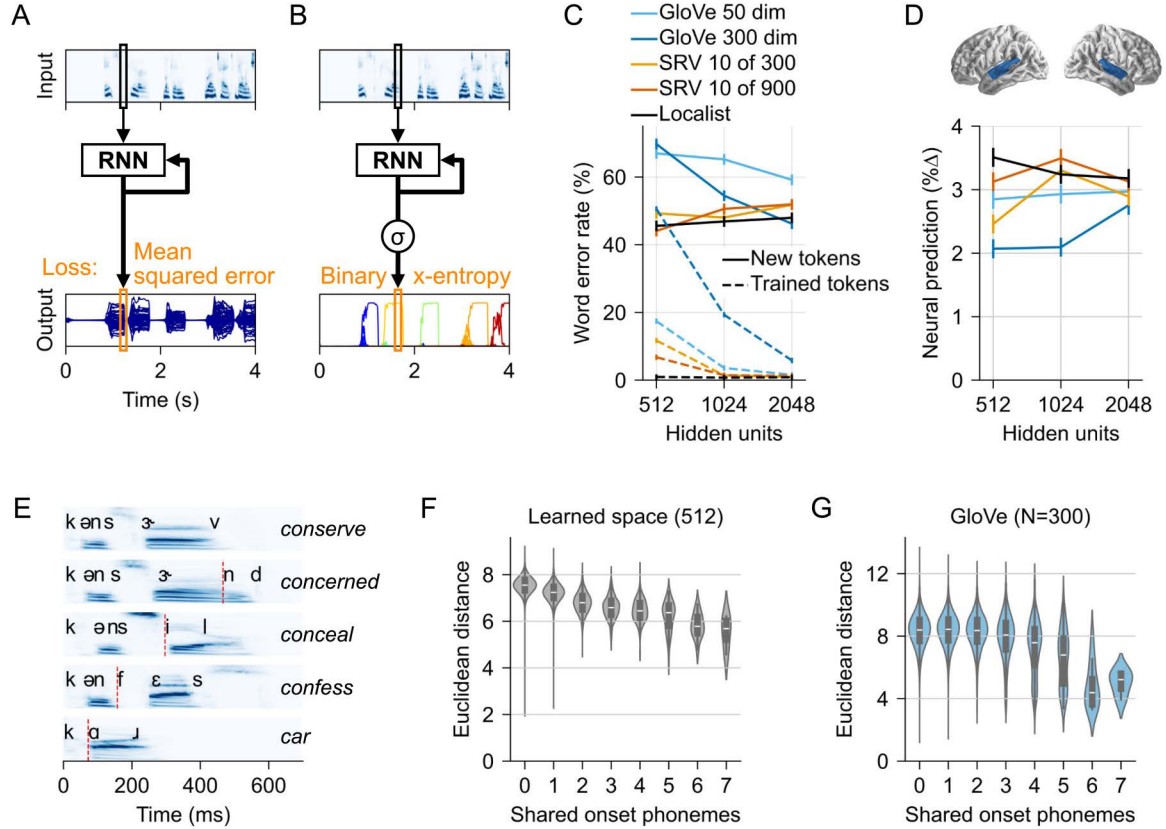

**Fig 2. Models for sparse output spaces better predict neural responses, and learn phonetic structure of the input lexicon.** (A) Architecture for RNNs with semantic output space. Word targets are dense vectors in a semantic vector space (GloVe). Each word has non-zero values in most or all output elements (blue lines). Output is evaluated using mean squared error (MSE). (B) RNN with sparse output, where targets are binary vectors, and each word is defined as 1 element (localist; shown) or 10 elements (sparse random vectors). In the illustration using a localist output space, each word leads to activation dominated by a single output element (different elements are distinguished by color). (C) Word error rate as a function of the numbers of hidden units, showing that all models can learn the task given enough hidden units. Error bars indicate within-subject standard error [34], treating each speaker as a subject (i.e., indicating the estimated standard error of the mean word error rate by speaker, calculated separately for the training and test sets) (D) Predictive power of each model for brain activity (quantified as % improvement in explained variance over an acoustic baseline model in the region of interest shaded in blue). All RNN models significantly add predictive power to the auditory-only model, but sparser RNN models have consistently higher predictive power than semantic vector space models. This difference seems to diminish with increasing number of hidden units. Error bars represent the within-subject standard error of the mean [34]. (E) Words commonly share the same acoustic-phonetic beginnings and can only be identified by considering all information across time. The plot illustrates this for *conserve* (first line): words lower in the graph share fewer word-initial phonemes. The point at which each word starts to differ from *conserve* is marked with a red line. (F) This acoustic-phonetic structure is reflected in the learned output mappings of localist models (dense layers): Words that share more onset phonemes are also located closer together in the output mappings. The graph shows the pairwise distance between words (x-axis) as a function of how many onset phonemes they share (y-axis). (G) Analogous analysis for the GloVe output space. The increased proximity of words sharing 4 + phonemes may be due to morphological structure (e.g., *story* and *stories* share onset phonemes because they have the same morphological root).

thus 10/300 matches the sparsity of the original, whereas 10/900 matches the target space size to the new lexicon size of 2934 words).

When given a large enough number of hidden units, RNNs were able to learn each of the output representations (Fig 2C), reaching ≤ 10% word error rate on trained tokens and ≤ 60% word error rate on novel tokens with 2048 hidden units. At lower numbers of hidden units, the denser output models performed notably worse.

The predecessor of the models reported here (EARSHOT) achieved a 33% word error rate for excluded words [19]. This model used 512 hidden units and a 10-out-of-300 SRV target space. Error rates in the comparable new models may

be higher (49% for 10-out-of-300 SRV and 44% for 10-out-of-900 SRV) due to the increased output lexicon size (2934 words vs 1000 words). Moreover, the new lexicon used here was constructed by deliberately including many phonological neighbors (word pairs that differ in only a single phoneme), which likely increased errors due to similar sounding words.

## Models with sparse output space predict human brain responses better than models with pre-structured semantic output space

Network performance did not strongly correspond to how well activity in the different RNNs predicted human MEG responses to speech. Fig 2D shows the predictive power of each model for brain activity localized to the superior temporal gyrus of both hemispheres. Predictive power was quantified as the improvement in explained variance from adding the RNN activity predictors to an auditory baseline model [35], i.e., 0% corresponds to the predictive power of the auditory model alone, and each data point in Fig 2D corresponds to the predictive power of a model combining the auditory model with the hidden unit activity from an RNN model. Note that while improvements may seem small, this is variance *uniquely* attributed to the RNN, and does not include variance that is shared between RNN and auditory predictors. Importantly, the improvements are highly reliable across subjects, as indicated by the error bars.

The models' predictive power for brain data varied by the number of hidden units and target space (*n-hidden* × *target-space*: $F_{(8,136)}=8.93$, $p<.001$). Overall, the pattern of results suggests that the main feature that makes a model a better predictor of brain activity is the sparsity of the target space, with sparser target spaces performing better than the denser spaces. The densest target was GloVe, in which each dimension varies for all words; SRVs are less dense, with only 10 non-zero values per word, while localist targets have exactly 1 non-zero value. Overall highest predictive power for brain data was associated with the 512 unit localist model (the only pairwise difference not significant was the one with the 1024 unit SRV models). Numerically, the sparser SRV (10 in 900) always outperformed the denser SRV (10 in 300; 512 units: $t_{(17)}=4.36$, $p<.001$; 1024: $t_{(17)}=1.73$, $p=.102$; 2048: $t_{(17)}=2.50$, $p=.023$). Sparser models also largely outperformed the semantic vectors matched in number of hidden units. The localist model was better than the GloVe-300 models at all levels (512: $t_{(17)}=5.97$, $p<.001$; 1024: $t_{(17)}=4.33$, $p<.001$; 2048: $t_{(17)}=2.42$, $p=.027$). An ANOVA including only the localist and GloVe-50 models also indicated a main effect of *target-space* (*n-hidden* × *target-space*: $F_{(2,34)}=2.23$, $p=.123$; *target-space*: $F_{(1,17)}=5.49$, $p=.023$), although pairwise tests were not significant at all levels (512: $t_{(17)}=3.10$, $p=.007$; 1024: $t_{(17)}=1.73$, $p=.102$; 2048: $t_{(17)}=0.86$, $p=.403$).

To verify that the selection of the region of interest did not bias these results we also performed whole brain tests comparing the GloVe-50 and localist output models' ability to predict brain signals. The GloVe-50 model did not predict brain signals better than the localist model in any brain region ($p\geq.84$, one-tailed, for 512, 1024 and 2048 hidden units). In contrast, the localist models consistently predicted brain signals better than the GloVe-50 models ($p\leq.006$, one-tailed, for 512, 1024 and 2048 hidden units). In sum, RNNs that were trained to map sound to abstract word forms developed temporal dynamics of activity that resembled human brain activity more closely than RNNs that were trained to map sound directly to semantic vectors.

## Localist models learn acoustic-phonetic space

The semantic GloVe space serves as a low dimensional embedding and thus imposes a pre-defined structure on the lexicon. In localist models, the dense mapping from RNN hidden units to the 2934 word vector can similarly be interpreted as an embedding, as it encodes the 2934 words in a lower dimensional space (the RNN output units). Thus, one reason the localist models develop more human-like temporal dynamics may be that they learn a lexical neighborhood structure that is more reflective of human perception.

During word recognition, humans initially activate multiple lexical candidates [9]. These early activations are determined by phonological similarity, as the first few phonemes of a word are usually compatible with multiple words. For example, upon hearing /bi/, listeners may consider both a beetle /bitəl/ and a beaker /bikɚ/. Fig 2E illustrates this for "conserve"

(top), with competitor words that share different numbers of onset phonemes (the red line indicates the point at which they start to differ from "conserve" based on a phonological segmentation). This co-activation of words is driven by the acoustic-phonetic overlap, and is independent of any semantic relationship between the words. However, a semantic space like GloVe constrains such co-activation by its internal structure [29]. For example, if *beaker* and *beetle* were on opposite sides of the space, they could not truly be both considered. The best the model could do would be an output corresponding to a point somewhere between the two words (which may incidentally correspond to a third, phonologically unrelated word).

We hypothesized that localist models may develop more human-like temporal dynamics by learning a lexical space that is structured such that it allows human-like co-activation of phonetic competitors. In such a space, words that are phonetic competitors ought to be closer to each other than words that are not. This was indeed borne out by the columns of the learned dense layer mapping RNN output to lexical identity. Fig 2F shows the average pairwise distance between words in this space for the 512 unit model, as a function of acoustic-phonetic overlap, quantified as the number of shared onset phonemes. The correlation between pairwise distance and phonological overlap was highly significant (given the large number of word pairs: $r = -.25$, $p < .001$) and was strong even among words with 4 or fewer shared onset phonemes ($r = -.25$, $p < .001$). Fig 2G shows the same analysis of the GloVe space for comparison. Here, the correlation is very low for 4 or fewer shared onset phonemes ($r = .00$) and only increases for words which share 5 or more onset phonemes ($r = -.24$). This latter increase is likely because words that share the same root are at the same time semantically related and phonological competitors (e.g., story/stories; breath/breathy; complain/complaint; vision/visual, truck/trucker, …).

**Depth makes semantic output models more human-like**

We next asked whether adding hierarchical depth affects the similarity between RNN activity and human brain activity. Hierarchical depth was introduced by adding multiple recurrent layers, i.e., the output of the first layer is the input for the second layer and so forth (Fig 3A). The flat model that proved most human-like in the preceding section served as a baseline: 512 hidden units with localist output. The RNN with 512 hidden units had 1,181,696 trainable parameters (excluding the output mapping). The number of RNN hidden units in deep models was determined by selecting the multiple of 64 that led to the number of trainable parameters closest to this (2*320: 1,313,280 parameters; 3*256: 1,379,328 parameters; 4*192: 1,084,416 parameters; note that while some of these are slightly higher than the flat model, flat models with a large number of parameters do not perform better, see above). The addition of hierarchy may be particularly important for GloVe models, as the addition of depth may allow more human-like acoustic-phonetic competition in lower levels, which are less directly constrained by the output space. Deep GloVe models might thus be better predictors of human brain activity by developing phonetic competition in lower layers, and semantic activation in higher layers.

In terms of word recognition, all models benefitted from depth, although the localist model consistently outperformed the GloVe models (Fig 3B). Initially, for predicting brain activity from the deep models, units were grouped by layer. Each layer of the model was used to create two predictors (cf. Fig 1). Using this methodology, deep models were more predictive of brain activity than flat models, and deeper GloVe models reached similar levels as the localist model (Fig 3C, solid lines). This result suggests that hierarchical layers provide meaningful subgroups of units, with different layers providing complementary information about human neural activity time courses. However, this analysis of the effect of depth also included a confound, because deep models provided more predictors than flat models. While flat models have no a priori groupings of units, such as the grouping into layers, they may nevertheless spontaneously develop different sub-populations of units with different response dynamics.

To account for this, we conducted a follow-up analysis in which we grouped units with *K*-means clustering. This method was equally applied to flat and deep models, i.e., in deep models the clustering algorithm was unaware which layer a unit belonged to. For the localist model, *K*-means clustering of the flat model led to consistently higher predictive power than unit-grouping of deep models by layer (Fig 3C, dashed black line). This suggests that flat localist models also developed

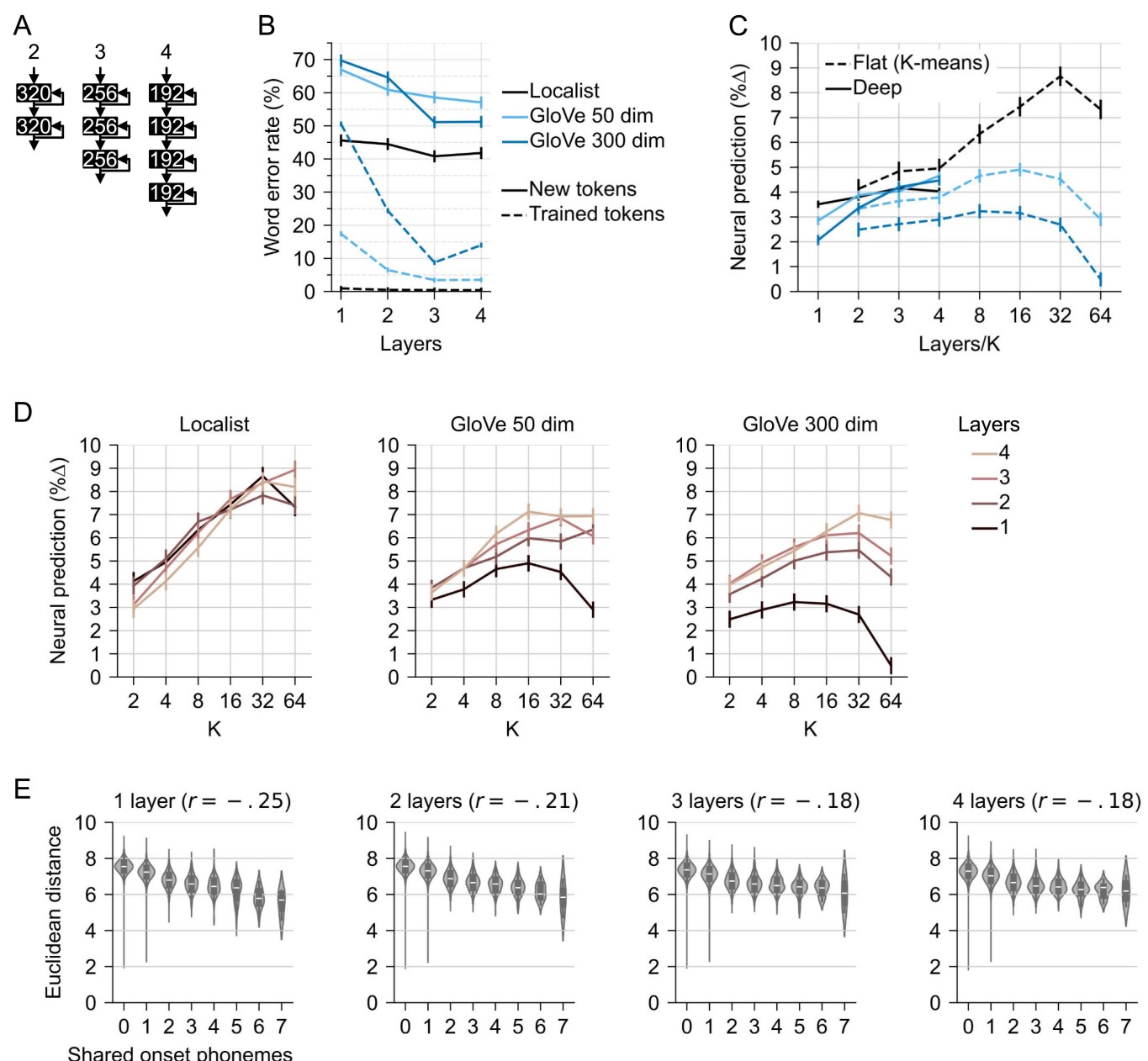

**Fig 3. Depth and unit sub-grouping improve performance and neural prediction.** (A) depth was added by stacking RNNs while controlling the number of trainable parameters (the number of hidden units per layer is indicated in white). (B) Deeper models have lower word error rate. (C) Deeper models also achieved better predictive power for brain responses, when units in each layer were used to create separate predictors (solid lines). However, increasing the number of predictors derived from flat models by K-means clustering also increased predictive power (dotted lines), with the highest predictive power for the flat localist model at K=32. (D) K-means clustering applied to deep models (ignoring the differentiation of units into different layers). While depth generally improved predictive power of GloVe models, it did not seem to have a clear effect on localist models. (E) Acoustic-phonetic structure in the learned output mappings is found even in deep localist models, but it decreases as a function of depth (analogous to Fig 2F).

meaningful sub-groups of activity patterns, as did hierarchical models. For GloVe models, $K$-means clustering still improved predictions of human neural data, although not as much as introducing hierarchy (Fig 3C, dashed blue lines). However, $K$-means clustering of the deep models led to a robust improvement over the flat models (Fig 3D). In contrast, for localist models, the effect of hierarchy was less clear. An ANOVA indicated a $K \times layers$ interaction ($F(15,255)=23.28$, $p<.001$), but the 1 layer, $K=32$ model was not significantly different from the 3 layer, $K=64$ model ($t(17)=.79$, $p=.44$). Selecting the best model in each family, the predictive power of the 1 layer localist model ($K=32$) was higher than the 4 layer GloVe-50 ($K=16$, $t(17)=3.21$, $p=.005$) as well as the 4 layer GloVe-300 model ($K=32$, $t(17)=5.10$, $p<.001$). Overall, this suggests that models develop meaningful sub-groups of units which are recoverable by $K$-means clustering. In GloVe output models, the addition of hierarchy leads to more human-like temporal dynamics. However, even the deep GloVe models do not match the predictive power of the localist models.

Deeper models separate acoustic input from lexical output, and output mapping may thus become less tied to the acoustic-phonetic structure of the input. On the other hand, the temporal nature of speech recognition may still favor co-activation of phonetic competitors. Acoustic-phonetic structure of the learned output space indeed decreased in deeper models, but only moderately, from $r=-.25$ in the 1 layer model to $r=-.18$ in the 4 layer model (Fig 3E).

## A modified loss function for human-like lexical competition

Fig 4A shows lexical competition in the output of the 1 layer localist model during presentation of the target words spoken by the human speaker. For each stimulus, the activation of all words in the output lexicon was recorded and sorted by the number of word-initial phonemes shared with the target (cf. Fig 2E). Comparing target and competitor activation (Fig 4A, left) suggests that while strong target activation reflects high recognition performance, this model exhibits little activation of competitors that share word-initial phonemes. Theoretically, target activation should occur later for targets with longer duration of competition. To visualize this directly, we computed relative target activation over time (target activation divided by the sum of activation of all words in the lexicon). The right panel in Fig 4A shows the time course of relative target activation as a function of the duration of competition (quantified as the number of onset phonemes shared with at least one other word in the lexicon). This confirms slightly steeper target activation for targets with fewer shared phonemes. However, high early target activation is seen even for words with long competitors. High relative target activation can only be achieved once the model de-activated all competitors. For comparison, Fig 4B shows a theoretical prediction of relative target activation, based on the phonetic transcriptions of the stimuli [36]. Here, target probability for each word was defined as 1/ the number of words in the cohort based on shared phonemes. Fig 4A suggests that this model identifies tokens earlier than should be possible based on phonetic input. This early preference for targets is also inconsistent with earlier work [37] using Elman networks [6] trained to activate word outputs (with a small lexicon) from abstract phonetic feature inputs presented phoneme-by-phoneme. This network generated activations that closely matched the conditional probabilities of words given the unfolding phoneme sequence.

The pattern of early target activation in responses to human tokens could indicate that the model identified these tokens based on token-specific acoustic details, rather than phonetic perception. Fig 4C shows the same plots for responses to synthetic speakers, suggesting that more realistic cohort competition developed in response to synthetic speakers, as seen in the original EARSHOT model [19], and perhaps reflecting lower acoustic distinctiveness of synthetic tokens. To further verify this, we trained a model with identical architecture, but with different data partitions, such that 1000 human speaker target tokens were never presented during training (i.e., only the synthetic recordings of those words were used during training). Responses aggregated over all 1000 untrained tokens reflected the increased word-initial phonetic ambiguity (Fig 4D, upper panels, with theoretical prediction for the subset of 1000 words in Fig 4E). However, examining trials on which the target was correctly recognized (26.8% of trials) revealed that, while target activation was delayed, the model activated few lexical competitors (Fig 4D, lower panels). In contrast, untrained synthetic talker tokens were associated with robust phonetic competition (Fig 4F). Thus, the model may have learned to pursue a different

PLOS Computational Biology

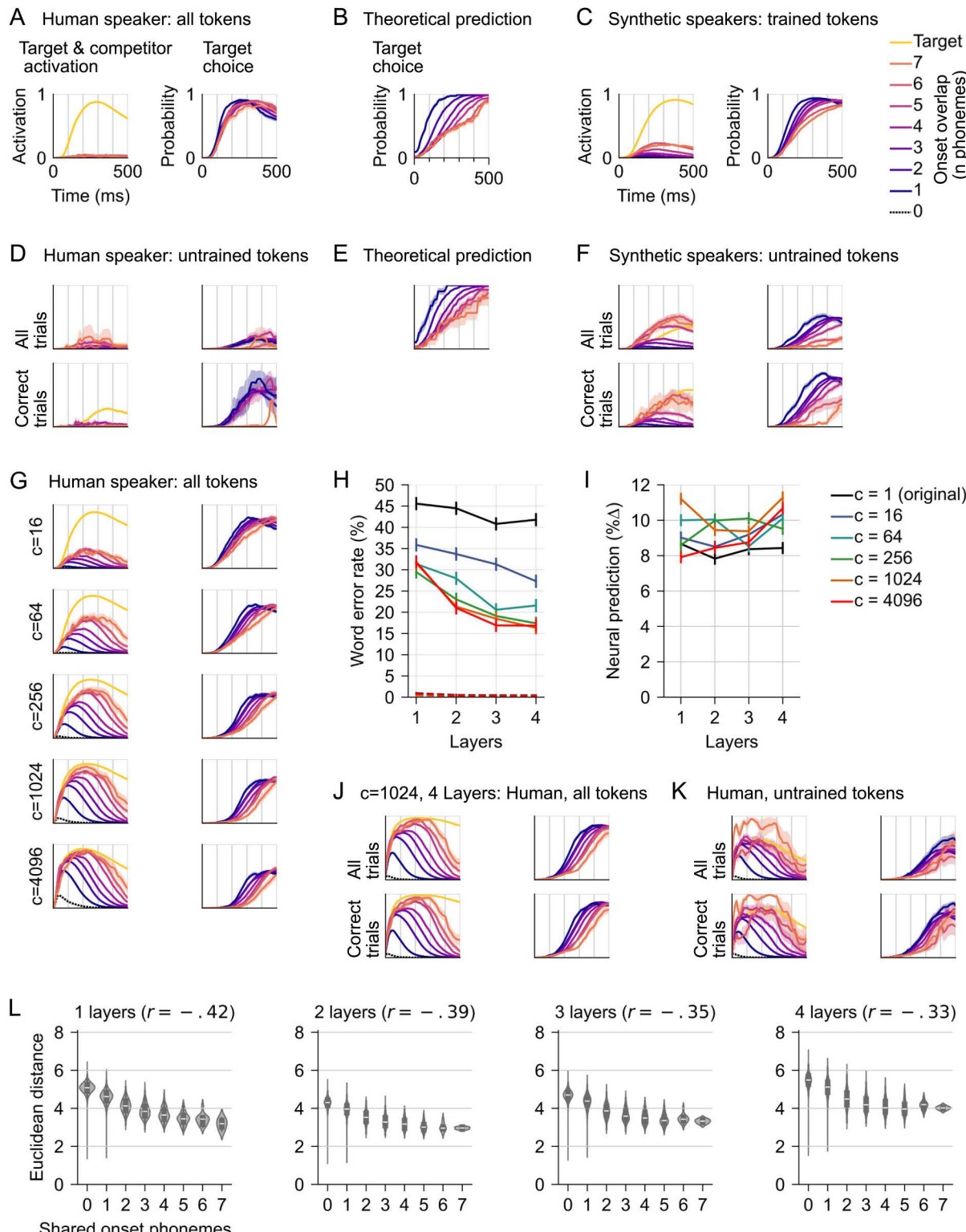

**Fig 4. Modified loss function that enables phonetic competition improves performance and neural prediction.** (A) Left panel: Activation of cohort competitors in response to human speaker words is limited (shown for the 1 layer localist model). The plot shows model output values in the localist space, interpreted as lexical "activation", as a function of time since word onset. Shown is the output for the target (yellow), and the average

for non-target words, with color indicating how many word-initial phonemes the respective non-target shares with the target. Only data for the human speaker tokens are shown (including 2751 trained and 183 untrained tokens). Values range from 0 to 1 due to the sigmoid activation function. Right panel: The relative activation of the correct target, as a function of how many word-initial phonemes the target shares with at least one other word (indicated by color). Relative activation is calculated as activation of the target, divided by the sum of the total activation in the output space, at each time point. Because the activation of the target is divided by the total activation, this can be read as a probability estimate for the target. Words that share more phonemes with competitors should take longer to reach high probability. (B) Theoretical prediction for target choice (A, right panel), based on phonetic transcriptions of the human speaker tokens. Target probability for each word was defined as 1/ the number of words in the cohort based on shared phonemes. (C) Same as A, but for synthetic talkers, only including tokens used in model training. Responses exhibit somewhat increased cohort competition. (D) A subset of 1000 human speaker target tokens that were never presented during training (in a separately trained model; data for the same 1000 tokens that were used in the MEG experiment). The upper panels show data for all trials. While target activation is delayed compared to A, lexical activation is generally low. The lower panel shows only correct trials (26.8%). High relative target activation suggests that the model pursued a similar strategy as for trained human tokens, activating a single candidate. (E) Analog to B but for the subset of 1000 words used in D. (F) Responses to synthetic tokens not used during training exhibit more normal competition effects than the responses to human tokens. This may indicate that synthetic tokens are acoustically more homogeneous. (G) The loss function was modified to reduce the penalty incurred by activating non-target words early during word presentation. Plots show responses to all human speaker tokens (including previously seen tokens, as in A). The modified loss function led to more human-like activation of cohort competitors early during human speaker words (left column), and delayed the point at which activation of the target exceeded other words, with the expected gradation by how many onset phonemes that competitors share with the target (right column), more consistent with acoustic ambiguity of the word onsets (data from 1 layer models). (H) The modified loss function led to improved WERs, and (I) substantially increased predictive power for human brain activity. (J) Lexical activation in a deep model was comparable to that of the 1 layer model. (K) The new loss function led to more realistic lexical activation of untrained human speaker tokens (same procedure as for D). (L) Acoustic-phonetic structure in the output mappings (c = 1024, analogous to Fig 2F).

strategy for human talkers than for synthetic talkers, exploiting the acoustic distinctiveness of human speaker tokens to identify a target as early as possible.

These results suggest that the acoustic variability in human tokens did influence model behavior, compared to training on entirely synthetic datasets [19]. This acoustic variability led the model to prioritize minor acoustic details that allowed identifying individual trained tokens as early as possible (Fig 4A), and increased the challenge in identifying untrained tokens (Fig 4D). In contrast to this model behavior, it is thought that humans activate multiple candidates that are compatible with temporarily ambiguous input [38]. Activating multiple candidates may be beneficial because it allows evaluating different possibilities, while deferring a final decision until it can be made under acoustic uncertainty. The model training schedule may have prioritized activating a single target as quickly as possible. This may be a consequence of the binary cross-entropy loss, applied throughout the duration of the word:

$$L_{bce} = -\frac{1}{N} \sum_{i}^{N} y_i \cdot \log\left(\hat{y}_i\right) + (1 - y_i) \cdot \log\left(1 - \hat{y}_i\right)$$

Where $N$ is the number of targets, $y_i$ the label (1 for the target, 0 for all other words) and $\hat{y}_i$ the model output (i.e., word $i$ activation). This loss function quantifies deviance from the labels without distinguishing between targets and non-targets. Intuitively, the loss equation implies that wrongly activating a single non-target affects the loss as much as activating the correct target. The loss is assessed independently for each output element, which is in principle consistent with multi-label classification during periods where the acoustic signal is consistent with multiple targets. When activations in the training signal sum to 1, the optimal prediction $\hat{y}_i$ can be interpreted as the probability of word $y_i$. This should allow such a model to learn probabilistic word activation reflecting lexical competition, as seems to be the case for synthetic speakers. However, the reward for activating only the correct target as early as possible is very high, which may encourage the model to make premature commitments (which may be suboptimal for more naturalistic input). Additionally, while probabilistic activation (i.e., $\sum_{i}^{N} \hat{y}_i = 1$) makes sense for a classifier, human cortical architecture is massively parallel, which may allow considering multiple words matching partial input simultaneously to a degree that exceeds this constraint.

To develop models that activate cohort competitors, the loss function was modified from the beginning of each word up to 100 ms before the end of the word. During this period, the loss for the target was left unchanged, but the loss for all other words was divided by a constant $c$ (Live and Let Live Loss, $L_{lill}$):

$$L_{lill} = -\frac{1}{N} \sum_i^N y_i \cdot \log(\hat{y}_i) + \frac{1}{c} (1 - y_i) \cdot \log(1 - \hat{y}_i)$$

The contribution to the total loss from non-targets (i.e., where $y_i = 0$) is divided by $c$. Intuitively, the model was still rewarded for activating the right target, but the punishment for activating non-target words was reduced.

Our variant of binary cross-entropy shares conceptual similarities with the focal loss [39] which, by focusing the loss on hard examples, indirectly downplays the contribution of non-target logits. It is also close in spirit to the asymmetric loss [40], which more explicitly weighs down non-target logits, although with a more sophisticated mechanism than the simple multiplicative constant we use here.

The new $L_{lill}$ loss function led to more human-like activation of phonetic competitors, even in trained-on tokens (Fig 4G, compare right-hand panels with Fig 4B). Models with high values started indiscriminately activating phonologically unrelated words (see $c = 4096$, dotted line with peak close to time 0, indicating activation of words that are phonetically unrelated to the target). The new loss function led to improved word recognition for previously unseen tokens, especially in combination with model depth (Fig 4H). Even at $c = 16$, all models performed better than their corresponding baseline with same depth but original loss ($t(15) \geq 6.31$, $p < .001$). The moderate levels of $c$ also generally increased the predictive power for brain activity in the STG (Fig 4I). When picking the best model for each level of $c$ (across number of layers), all modified loss models performed better than constant loss ($t(17) \geq 2.66$, $p \leq .016$; all for $K = 32$). This result suggests that models with more human-like lexical competition also exhibit information processing dynamics that correspond more closely to human brain dynamics. The effect of model depth was less clear: at $c = 1024$, predictive power of the 1 and 4 layer models was statistically indistinguishable ($t(17) = -0.17$, $p = .86$). Model depth also did not notably affect cohort competition behavior (Fig 4J). Models with modified loss also exhibited more realistic target activation patterns on untrained human speaker tokens (Fig 4K). Finally, increased cohort competition was also reflected in the acoustic-phonetic structure of the learned output space (Fig 4L).

### The RNN explains more variance than phoneme surprisal

The 1 layer, $c = 1024$ model improved predictions of MEG data by 11.2% over the auditory baseline model in the region of interest (Fig 4H). We have previously found that phonetic surprisal is a significant predictor in the same dataset [41]. For comparison, phoneme onsets derived from forced alignment improved predictions by 0.2% over the baseline model, and phoneme surprisal improved them by another 0.4%. Cohort entropy, which is a significant predictor of brain responses to continuous speech [10,42], was not significant in this isolated word experiment [41]. Overall, this suggests that the effect found for the RNN is not easily reduced to known psycholinguistic variables.

### Discussion

RNNs have been proposed as tools to help us understand the computational demands for human cognitive processes [6], including speech recognition [8,19]. Our results demonstrate that this relationship is not just conceptual. RNNs trained to recognize spoken words indeed develop temporal dynamics in their operations that predict human cortical responses to the same words. This suggests that operations learned by those RNNs are in some respects reflective of computations performed for word recognition in the human brain. Critically, altering the details of the task learned by the RNNs alters the predictive power of those RNNs for human brain activity, such that RNNs that are designed to be more consistent with

psycholinguistic principles are also more predictive of brain activity. These results suggest that RNNs indeed learn computations that resemble information processing in the human brain.

### Why might RNN activity predict brain activity?

Artificial neural networks, although originally inspired by biology, emulate biological principles very selectively. LSTMs were designed to solve a computational problem, not to emulate how biological networks work [12]. Nevertheless, hidden unit activity in LSTMs trained to recognize speech can predict the temporal dynamics of neural activity in human listeners. Why might this be? The values in LSTM units store the result of local computations, and make those results available to other units. LSTM units thus transmit information, both forward in time, and to the RNN's output, or the next layer in hierarchical models. Similarly, human neural activity transmits information within and across regions. Specifically, activity in dendritic trees, the principal generators of the MEG signal, is related to integrating information that is received from other neurons. Thus, the affinity between RNNs and human brain activity may indicate that the two systems develop similar information processing strategies when faced with the same speech signals, and, consequently, that they update internal representations at similar points in time, which leads to similar temporal dynamics in information transmission. This line of reasoning is reinforced by the results showing that when computational demands for RNNs are made more consistent with human cognition, the RNNs also develop more human-like activity patterns. Finally, LSTM internal connectivity may resemble cortical microcircuits [13], which may have further promoted similar information processing strategies.

### Neural network activity as a linking hypothesis

A neural linking hypothesis specifies which quantifiable aspects of a cognitive model predict brain activation. In research predicting language-related brain activity from neural network models, typically the values of all hidden units in a specific network layer are used to predict neural population activity [27,28]. Here we introduced a novel linking hypothesis, by averaging activation across groups of units. This linking hypothesis is motivated by the physiological basis of M/EEG signals. The electrical activity produced by individual neurons is not strong enough to be measurable above noise levels with M/EEG, but these signals reflect simultaneous activity in populations of spatially aligned cells, i.e., activity in many current vectors that is averaged physiologically [43]. Even though units in artificial neural networks do not necessarily correspond to biological cells, we hypothesize that our artificial networks developed population dynamics that can be related to biological neural population dynamics. Thus, on this approach, we treat the neural network more like a subject in an electrophysiological experiment, rather than assuming arbitrary access to each individual unit's activity.

   The advantage of the linking hypothesis employed here is that 1) all hidden units of the RNN are incorporated, not just a single layer, and 2) the hidden units within a functional group (layer or *K*-means cluster) are all assigned the same weight for predicting brain activity. The latter is important because in a regression model in which each hidden unit is assigned separate regression weights, the model can selectively use units that resemble brain activity and ignore ones that do not. This is problematic because in such high dimensional models, effects may be driven by a relatively small number of units that happen to resemble human responses by chance. For instance, even untrained deep networks, initialized with random weights, can make surprisingly accurate brain activity predictions [22,44]. This is likely because untrained but structured deep networks provide rich reservoirs of features [45,46], some with response characteristics resembling neural processes *by chance*.

### Lexical activation and competition

During word recognition, human listeners start interpreting partial input early and consider multiple lexical candidates for recognition. For example, upon hearing /bi/, listeners activate lexical representations for *beaker* and *beetle* [9,47,48].

This is typically conceptualized with a model of the mental lexicon in which each word can be independently activated [38], as is the case in a localist semantic space, where activation of any one word is orthogonal to other words. In a lower-dimensional space, in which words are not orthogonal, the space constrains which words can be distinctly activated simultaneously [29,49]. In a semantically structured space such as GloVe, activating a word entails activating other words in its semantic neighborhood (e.g., the words closest to *valley* in our GloVe-300 space are *hills*, *area*, …). However, words that are phonetic onset competitors are not clustered systematically (Fig 2G). To implement phonetic competition, such a model may point to a location corresponding to an average of the competitors [49]. However, this average location may be closer to phonetically unrelated words than to any of the possible targets. For instance, in humans, /væl/ may activate *valley*, *value*, *valet* and *valid*, but in our GloVe-300 space the words closest to the arithmetic mean of these four vectors are *hence*, *example*, and *fact*.

Our results suggest that RNN architectures that allow and facilitate simultaneous activation of phonetic competitors develop the most human-like temporal dynamics. First, models with a trainable target space structured that space according to acoustic-phonetic principles. This may allow them to process the input more efficiently over time by directing their lexical search towards neighborhoods in the target space that contain words with similar onsets. Second, a modified loss function that reduced punishment for activating multiple candidates during early stages of word recognition led to even more human-like temporal dynamics in LSTM units. The new loss function may allow for more unconstrained exploration of the lexical space during early stages of word recognition, precluding premature commitments to specific words. In contrast, models that directly mapped sound to the semantic GloVe space made more errors and developed less human-like temporal dynamics. Our results, together with the theoretical considerations above, suggest that a phonetically structured target space is a more appropriate model for human word recognition than a direct mapping to a semantically structures space.

This is not to say that semantic activation is an afterthought during human word recognition. Semantic activation may occur even before a word can be identified [47,48]. For example, hearing *logs* primes *key*, because *logs* temporarily activates the phonetic competitor *lock,* which is a semantic associate of *key* [47]. However, a single vector in a semantic space is also a poor model of such semantic priming, for the same reason indicated above: it would not allow selectively priming associates of two phonetic competitors without also priming all the concepts that lie in the space between. To account for such facts, a semantic model should allow multiple entry points, for instance multiple vectors in a semantic space rather than a single vector (/væl/ primes *mountain* through *valley*, and *parking* through *valet*, but not all the words that lie in between those, like *example* and *fact*). Models with localist outputs performed well because the localist space provides the right kind of top-down constraint on lexical co-activation patterns, and enables information processing dynamics corresponding to phonetic competition. Localizing semantic *representations*, as opposed to temporal dynamics of word recognition, is a separate problem. For now, note that the lack of semantic representations in our localist model gives it a similar scope (mapping from inputs to word forms) as TRACE [50] or Shortlist B [51], but with the major advance over those models that it operates on real speech inputs.

**Loss function for word recognition**

The loss function employed here is designed to produce lexical competition, rather than as a developmentally realistic learning model. Human word recognition is probably more driven by the need for context-appropriate lexical interpretations. Nevertheless, lexical activation during word recognition, before a unique lexical item can be identified, can be characterized as a probabilistic problem, i.e., the perceiver may generate a probability distribution over what word they are currently hearing [37,51]. This account motivates the binary cross-entropy loss function, which we initially used for localist models, because the binary cross-entropy can be minimized by setting each output element to its probability. However, when used as a model for brain activity, this may implicitly assume that neural activation of lexical candidates also ought

to sum to 1. In our case, where each word has the same lexical frequency, if a word onset is consistent with 10 words, each of those lexical representations should be activated at 1/10. This may not be a good characterization of how the brain activates lexical candidates. In fact, brain activity has been found to correlate with the entropy of lexical distributions [42,52]. We found that a loss function that relaxed the constraint, and allowed the model to more fully activate candidates, led to activity patterns that were more consistent with human brain activity. This suggests that a model that can activate lexical candidates beyond their contextual probability may be a better model for human word recognition. This does not preclude the possibility that probabilistic considerations enter lexical decisions in other ways [51]. Even though lexical activation distributions produced by the new loss function do not sum to 1, a probability estimate of each candidate is implicit in the distribution and can be obtained by normalizing it.

An alternative (not mutually exclusive) interpretation of the success of the modified loss function centers on how the loss function may influence model strategy. Our localist model trained with the original binary cross-entropy loss exhibited a distorted pattern of lexical activation for trained-on human speaker tokens, favoring quick activation of the target before the phonetic recognition point (Fig 4A). This was not the case for trained-on synthetic talker tokens (Fig 4C) or untrained human speaker tokens (Fig 4D). Realistic competition also emerged in the original model EARSHOT, which used only synthetic speakers [19]. This indicates that our model may have learned to recognize trained-on human speaker tokens through memorizing token-specific acoustic details. Synthetic tokens may lack such token-specific acoustic idiosyncrasies. Our modified loss function reduced the disproportionality high reward for recognizing a word early on, which may have led the model to pursue an alternative strategy based on phonology rather than memorizing acoustic details. Interestingly, human speech may contain acoustic features that distinguish words with phonologically identical word onsets [53]. For example, vowel duration can distinguish *cat* from the otherwise phonologically identical onset of *catalog*. However, such non-phonetic acoustic cues are context-dependent [53], and relying on them for word recognition may be a suboptimal strategy for human listeners, which have to deal with more variable contexts.

## Hierarchy and information flow

We implemented hierarchy through the common technique of stacking LSTM layers. This architecture does not allow information to flow from later to earlier layers, preventing simulation of long-range feedback connections. The architectures may thus implement primarily feedforward information flow in speech recognition [51], although networks with a recurrent layer anywhere within their architecture violate the principles of autonomous architectures [54] because they are by definition cyclic graphs with a feedback loop: the bottom-up input to a recurrent layer is immediately inextricably integrated with top-down outcomes from processing the recurrent layer's previous states [55]. Recurrent connections in stacked LSTM models may simulate local feedback within layers, consistent with models of perception that emphasize local feedback connections [56]. However, the architecture may not be able to account for some effects hypothesized to rely on long range feedback connections. For example, ambiguous phonemes can sometimes be disambiguated by lexical knowledge (*Christma#* [with final sound replaced by noise] is heard as *Christmas* because there is no other word matching the pattern). A feed-forward architecture may reflect this by finding the appropriate output despite the noisy input, but psychophysical data suggest that humans also revise lower level acoustic-phonetic representations [57,58]. While our models developed temporal dynamics consistent with human brain activity, the addition of such feedback connections may be more important when input is noisy [59].

## Conclusions

Our results demonstrate that using RNNs as models of human speech recognition can provide a middle ground between high-level cognitive models and explicit neural implementations. These models learn brain-like computational principles using relatively simple and broadly available computational methods. The success of the LSTM architecture in our investigation adds support for the hypothesis that this architecture implements brain-like computations [13].

## Materials and Methods

### Recurrent neural networks

**Training.**  The stimuli were based on a set of 1000 target words used in a MEG experiment [41], spoken by a human male speaker for the massive auditory lexical decision (MALD) database [36]. To simulate realistic lexical neighborhoods for those target words, additional words were added to the lexicon if they 1) differed from a target word by a single phoneme, 2) had a frequency count of at least 1000 in the SUBTLEX-US corpus [60], and 3) were included in the MALD database. 51 homophones were removed. This resulted in a total of 2934 words. For each of these words, 15 additional tokens were produced using Apple "say" speakers (Agnes, Allison, Bruce, Junior, Princess, Samantha, Tom, Victoria, Alex, Ava, Fred, Kathy, Ralph, Susan, Vicki). All tokens were transformed to 64 band gammatone spectrograms, roughly corresponding to the frequency resolving power of human hearing [61] and allowing visually smooth formant movements (see Fig 1). Model training and analysis was performed at 100 Hz temporal resolution.

For RNN training, tokens were divided into training and test sets. For each of the 16 speakers, 1/16th of the tokens were randomly assigned to the test set, such that each word was trained with 15 tokens from 15 different speakers, and with one token/speaker held out for testing. To minimize item variability when predicting MEG data from RNNs, the randomization was constrained to always include the MALD tokens presented in the MEG experiment in the training set.

Training input consisted of a continuous stream of words and silence. The following rules were used to mimic the continuity in the input due to speakers commonly producing sentences rather than isolated words: 1) select a speaker; then, produce two phrases according to: 2) select a token from the speaker; 3) with a 50% chance, go back to (2); 4) insert a silence of 200–500 ms. Return to (1).

All RNN models were trained with TensorFlow 2.8 [62]. Inputs were created by concatenating gammatone spectrograms and silence into 10 s long segments, but implemented a fully continuous time axis: for example, a word might start at the end of one segment and continue in the beginning of the next segment, and LSTM states carried over across segments. Gaussian noise was added to all stimuli at 20 dB signal-to-noise ratio (determined separately for each frequency band). Models were trained in parallel with a batch size of 32. Each training epoch consisted of 25 10 s segments. Models were trained to minimize binary cross-entropy (localist and SRV models) or mean squared error (GloVe models) across *all* time points. Training loss was evaluated after each epoch. When the loss increased over a previous minimum for 20 epochs, model weights were restored to that previous minimum. When the loss failed to improve for 200 samples, model training was terminated.

**Evaluation.**  For model evaluation, the trained models were presented with a continuous sequence of all tokens without any intervening silence, and without added noise. Training set tokens were presented before the test set tokens; otherwise, the order was completely randomized (without replacement). To evaluate model performance, each model's output was transformed into lexical activation (i.e., a vector assigning a value to each word, where the largest value can be interpreted as the word that the model is currently perceiving). Localist models directly output a vector of activation for each possible word. For semantic vector space models, word activation was defined as the negative distance of the output from each word in the lexicon (i.e., the model's percept was defined as the word in the lexicon with the smallest distance from the output). For SRV models, the output was transformed into lexical activation by assigning to each word an activation value corresponding to the minimum across the 10 output elements corresponding to that word. Word error rate was then determined based on the average word activation (as defined above) during the last 100 ms (i.e., 10 samples) of the target word. The word was recognized correctly only if the most highly activated word corresponded to the actual target. For statistical evaluation, word error rate was calculated separately for each speaker.

### MEG dataset and analysis

MEG data were drawn from a public dataset [63]. Preprocessing and source localization are summarized here, and are described in more detail in a previous publication [41]. The analysis included data from 18 right-handed, native speakers

of English. Participants listened to 1000 words taken from the MALD database [36], with an interstimulus interval of 276 ms. Occasionally, a visual probe word appeared on the screen and participants judged whether the word was semantically related to the word they had heard most recently, and all participants included in the analysis answered at least 69% of probes correctly.

**Preprocessing.** Extraneous artifacts were automatically removed using temporal signal space separation [64]. Data were then band-pass filtered between 1 and 40 Hz, and biological artifacts were removed using extended infomax independent component analysis [65]. Data were then low-pass filtered at 20 Hz and downsampled to 50 Hz.

A source space for MEG analysis was defined on the white matter surface of the FSAverage template brain [66]. This source space was uniformly scaled for each participant to approximate their head size. MEG data for the whole experiment were projected to virtual current sources, oriented perpendicularly to the cortical surface, using regularized minimum $\ell 2$ norm estimates ($\lambda = 1/6$).

**Auditory baseline model.** A model of auditory processing was generated based on gammatone spectrograms as in previous experiments [35,41]. First, high resolution gammatone spectrograms were generated for all word stimuli with 256 frequency bands in equivalent rectangular bandwidth (ERB) space between 20 and 5000 Hz. These spectrograms were resampled to 1000 Hz and log-transformed to better approximate response characteristics of the auditory system [67]. Onset spectrograms were generated by applying a neurally inspired edge detector algorithm [3,35]. Spectrograms and onset spectrograms were downsampled to 8 frequency bins, equally spaced in ERB space, for mTRF model estimation.

In addition to auditory predictors, the baseline model also included a predictor with a single impulse at each word onset, to account for any generic onset processing not captured by the acoustic predictors. Thus, the baseline model contained 17 predictors: an 8-band spectrogram, an 8-band onset spectrogram, and 1 word onset predictor.

**RNN linking hypothesis.** The basic linking hypothesis we employed assumes that non-zero values in LSTM hidden units correspond to neural activity. To implement this, a predictor variable was defined as the sum of the absolute values of all hidden units in a layer, at each time point. A variation of this linking hypothesis assumes that biological neural activity may correspond to updates in RNN hidden units, rather than raw values [68]. To implement this, a second predictor variable was computed by taking the derivative in time of the absolute values of each unit, applying half-wave rectification, and then summing across units. Half-wave rectification, i.e., clipping negative derivative values to 0, assumes that hidden units moving away from zero represent new information, whereas returning to zero constitutes a return to baseline (compare hidden unit activities in Fig 1). Pretests suggested that both predictor types contributed independently to predictive performance of the models. In the reported tests we always include both predictor types, i.e., each LSTM layer was always represented by two predictor variables, reflecting magnitude and change of hidden unit activity.

In deep models, these predictors were computed separately for each layer. To test for meaningful unit groups in flat models, $K$-means clustering [69] was applied to the model activity to identify unit groupings with similar time courses. To explore how many meaningful unit subgroups contribute to MEG signals, $K$ was increased from 2 in powers of 2, up to 64, which generally showed a decrease in predictive power for MEG data compared to 32 (indicative of overfitting). For deep models, clustering was applied to all units together, i.e., the clustering algorithm was unaware of the layer each unit belonged to. In the best model (1 layer, $c = 1024$, $K = 32$), 32.0% of the variance predicted by the RNN could be uniquely attributed to the magnitude predictors ($t(17)=5.78$, $p < .001$), and 30.4% to the change predictors ($t(17)=7.66$, $p < .001$).

**Multivariate temporal response function (mTRF) analysis.** Source localized MEG responses were analyzed using multivariate temporal response functions (mTRFs) with Eelbrain [67]. Separate mTRF models were estimated for each set of predictors and for each participant. Each dataset (MEG data and time-aligned predictors for one subject) was analyzed as one continuous recording. First, the data were divided in 4 contiguous segments for 4-fold cross-validation. For each of the 4 test segments, the three remaining segments were used to estimate 3 different mTRF models, with each segment serving as validation segment once and as training segment in the other two runs.

mTRFs were estimated with a mass-univariate approach, independently estimating mTRFs for each virtual current source in the neural source space. In general, an mTRF model estimates the MEG signal time series at a virtual current source, $\hat{y}_t$, as a linear convolution of $N$ predictor variable time series, $x_{i,t}$, with the mTRF $h_{i,\tau}$, where $\tau$ indexes one of $T$ time delays between predictor variable and brain response:

$$\hat{y}_t = \sum_{i}^{N} \sum_{\tau}^{T} h_{i,\tau} \cdot x_{i,t-\tau}$$

In the current analysis, $\tau$ always ranged from -100–1000 ms. Within each run, a "boosting" coordinate descent algorithm was used to minimize the $\ell 2$ error: First, the training data segments were used to find the mTRF element $h_{i,\tau}$ which, when changed by a fixed step $\Delta$, led to the smallest error in the training data. Then, the validation segment was used to test whether the new mTRF also led to an improvement in the validation data. If the new change led to an increase in validation error, then $h_{i,\tau}$ was reset to its previous value, and the mTRF components for predictor $i$ were frozen. This proceeded until the whole mTRF was frozen. This regularization using a validation set minimizes overfitting by promoting sparsity in response functions [70].

The 3 mTRFs for each test segment were averaged and used to predict the MEG responses in the test segment. The test segments were then concatenated to compute the fit metric (% of the variance explained) for the whole dataset.

## Statistics

Statistics are implemented in Eelbrain [67]. For region of interest (ROI) tests, an ROI was defined as the posterior 2/3 of the combination of the anatomical labels for the superior temporal gyrus and the transverse temporal gyrus in each hemisphere [71], the same ROI as in previous work [10,41]. The % variance explained was averaged in this ROI and analyzed with standard univariate tests (related measures $t$-tests and ANOVA). For whole-brain mass-univariate $t$-tests, family-wise error correction was implemented with threshold-free cluster enhancement [72], with a null distribution estimated from 10,000 random permutations of the condition labels.

## Acknowledgments

We thank Kevin Brown, Emily Myers and Phoebe Gaston for discussions on related topics.

## Author contributions

**Conceptualization:** Christian Brodbeck, Thomas Hannagan, James S. Magnuson.

**Data curation:** Christian Brodbeck, James S. Magnuson.

**Formal analysis:** Christian Brodbeck.

**Funding acquisition:** Christian Brodbeck, James S. Magnuson.

**Investigation:** Christian Brodbeck, Thomas Hannagan, James S. Magnuson.

**Methodology:** Christian Brodbeck, Thomas Hannagan, James S. Magnuson.

**Project administration:** Christian Brodbeck, James S. Magnuson.

**Resources:** Christian Brodbeck, James S. Magnuson.

**Software:** Christian Brodbeck.

**Validation:** Christian Brodbeck.

**Visualization:** Christian Brodbeck.

**Writing – original draft:** Christian Brodbeck, Thomas Hannagan, James S. Magnuson.

**Writing – review & editing:** Christian Brodbeck, Thomas Hannagan, James S. Magnuson.

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
