## [Decision Letter · Decision Letter 0]

Dear Dr. Brodbeck,

Thank you very much for submitting your manuscript "Recurrent neural networks as neuro-computational models of human speech recognition" for consideration at PLOS Computational Biology.

As with all papers reviewed by the journal, your manuscript was reviewed by members of the editorial board and by several independent reviewers. In light of the reviews (below this email), we would like to invite the resubmission of a significantly-revised version that takes into account the reviewers' comments.

Dear Author,

The two reviewers are quite enthusiastic about your contribution but have list a comments that I would like you to address. Many of these are points of clarity but some are more significant. They are going to strengthen your manuscript.

Best,

Frederic Theunissen

We cannot make any decision about publication until we have seen the revised manuscript and your response to the reviewers' comments. Your revised manuscript is also likely to be sent to reviewers for further evaluation.

Sincerely,

Frédéric E. Theunissen

Academic Editor

PLOS Computational Biology

Thomas Serre

Section Editor

PLOS Computational Biology

Dear Author,

The two reviewers are quite enthusiastic about your contribution but have list a comments that I would like you to address. Many of these are points of clarity but some are more significant. They are going to strengthen your manuscript.

Best,

Frederic Theunissen

Reviewer's Responses to Questions

**Comments to the Authors:**

Reviewer #1: In this paper, Brodbeck et al. explore how RNNs, particularly LSTMs, can be used to model and understand the neural processes involved in human speech recognition. They trained LSTM RNNs to recognize speech from auditory spectrograms and tested whether these networks' internal dynamics can predict human neural responses to the same stimuli. They found that the RNNs' predictions of neural responses improved when the network architectures were modified to include cognitive principles like phonetic competition, suggesting a more human-like processing of speech. Furthermore, they demonstrated that RNNs can replicate some aspects of the hierarchical nature of neural processing in the brain, with different layers of the network modeling different levels of brain response to speech.

Overall, the paper is well-written and logically structured. I do have several comments and suggestions that could potentially strengthen the manuscript:

1. Temporal Dynamics and Phonetic Competition

The manuscript claims that modifications in the LSTM loss function enhance the model’s capability to capture phonetic competition, leading to better neural response predictions (as shown in Fig 3). However, it remains unclear if the additional variance explained by the new model with the modified loss function directly results from phonetic competition. The paper does not empirically demonstrate the existence of phonetic/lexical competition in the MEG activity. It is also not clear the emerged competition effect in the LSTM hidden units can be correlated to these MEG components. Without isolating the effect of phonetic competition from other factors, it might be premature to attribute improvements in prediction accuracy solely to this aspect.

2. Model Complexity and Prediction Accuracy

Figure 2G can also be interpreted as increasing the number of features in the mTRF model enhances brain prediction accuracy. This observation raises a question whether clustering the units into more clusters based on their activation profiles would lead to better neural prediction as well, since the grand averaging across all units may attenuate the between-unit variance. The extreme case would be to use activation of each unit as a distinct predictor, instead of averaging across units.

Another concern I have is that the heavily unbalanced number of predictors. The baseline have 256 features, while the RNN predictors only have 2 features. Would increasing the RNN predictors change the results?

3. Extension to Longer Linguistic Contexts

The current focus of this manuscript is primarily on single-word perception, which relies heavily on model training specifics (as noted in Fig 2C). Given RNNs’ capability to capture long-term dynamics, it would be valuable to see how this model performs with longer linguistic contexts, such as sentences. For instance, could pre-training the model on a task like automatic speech recognition or using pre-trained models like Deep Speech 2 influence the emergence of dynamics similar to those depicted in Fig 3I?

Minor Points:

1. The labeling of panels ABCDE in Figure 1 is somewhat confusing. Consider reordering these labels to reflect a top-to-bottom, left-to-right progression to enhance clarity.

2. In Figure 1, the feature is described as the summation over the absolute values of hidden unit activations, yet the absolute term appears to be omitted in the graphical representation. Clarifying this in the figure could prevent potential misunderstandings.

3. Is there any regularization included in the mTRF model? Given the dimensionality of the regression problem, it may be worth considering some sort of regularization, such as L2. I am also curious why the authors used coordinate descent as the solver for the regression.

Reviewer #2: I'm pleased to provide a review of this very interesting paper. The paper is well written, clearly organised and addresses a topical and important question in advancing our understanding of the neural and computational mechanisms that support perception and identification of spoken words by human listeners. This is a domain in which engineering systems have achieved considerable success - achieving near-human levels of accuracy even in challenging listening situations - by applying modern neural network methods. However, human-level performance need not imply human-like mechanisms and this paper therefore explores the architectural, representational and computational assumptions required for a recurrent neural network system to show human-like behaviour and neural activity.

The paper achieves some success in explaining both behavioural and neural data and thus makes a very useful contribution to this area of computational biology. However I had some comments on the relationship between the computational simulations and behavioural and neural outcomes which I think would be best addressed by revisions to the manuscript and further consideration by the authors. I'll discuss these two areas in turn, before adding some minor comments about specific points of presentation.

(A) Relation to behavioural data

(1) The first indicator of human-level behaviour are low word-error rates (WER). These are typically very low for skilled listeners (5-10% or fewer, depending on listening situations, stimulus materials and task/scoring criteria). Established DNN models already achieve similarly low error rates (e.g. Kell et al, 2018 Neuron). Interestingly, the model explored in the present work did not previously achieve such low error rates (this is the EarShot model from Magnuson et al, 2020, for which WER was around 30% - though this depends on the test and generalisation conditions that are considered most comparable). I think it would be useful for the authors to make this comparison with the previous version of Earshot more explicit. Is their model substantially different from and better performing than previous versions of EarShot? Which of the previously observed WER are most comparable to this new model? How does the change from the purely-synthetic speech used previously to the mixed natural and synthetic speech on which this model is trained impact WER and other measures? I think the authors should explain why this model is (seemingly) so much more accurate.

(2) One of the most interesting aspects of the original Earshot model, that is shared by this updated model, is that - while working with neurally-plausible temporal processing mechanisms (i.e. sequential audio input, and recurrent connections) - it accurately simulates the time-course of word identification, capturing well-established behavioural findings concerning the time-course of recognition of onset-aligned "cohort" competitor words (such as beatle and beaker). In the previous EarShot paper (Magnuson, 2020) this property was demonstrated in the main simulation, but changes to the input representation prevented this being observed (Supplementary Figure S1.3 using MFCC input). The present paper reports similarly variable findings in Figure 3D-F. However, it's not clear that these figures sufficiently justify the authors statement that "activation of the target was delayed reflecting the word-initial acoustic ambiguity".

It seems that the model is recognising target words at around 300ms, whereas the time-course of the input for words like conserve/concerned (in Figure 2H) would suggest that this is still rather early given the uniqueness points of some of the spoken words. While the depiction of the time-courses in Figure 2I are more persuasive, and the figure style used is impressively compact, I think that a more direct assessment and clearer depiction of the relationship between stimulus timing, stimulus uniqueness points and model recognition points would be helpful. Is there statistical evidence that recognition is time-locked to word uniqueness point rather than word offset in these simulations? Will these models capture differential effects of uniqueness point on recognition of words that are higher vs lower frequency members of their cohorts (suggested, for instance, by Dahan et al, 2001 or Gaskell & Marslen-Wilson, 2002)? My understanding of the literature is that these forms of behaviour are well-approximated by models that activate words in proportion to their conditional probability given the speech input heard thus far (in line with Norris & McQueen, 2008). Does the current model achieve this?

(3) In their later simulations the authors separate two aspects of the typically-used binary, cross-entropy error term by adding an additional parameter ("c") which reduces the penalty faced by the model for activating competitor words early on in the identification of specific spoken words. This achieves the desired outcome by (I think) relaxing the assumption that the model is performing multi-class classification (i.e. 1/N words are present at a time), and making the model perform something more similar to multi-label classification (i.e. several words may be simultaneously present and hence co-activated). This seems like the wrong assumption - the speech signal contains only one word at a time - but perhaps I've made a mistake in my interpretation here (I should note that one of my colleagues who co-reviewed didn't agree with my interpretation). Nonetheless, I had some concerns with the presentation and interpretation of this result, perhaps due to some very compact graphs and explanation, and perhaps also due to some problems with the behavioural and computational interpretation. Specifically:

- in the main text and in figure 3I there's inconsistency about whether c is used as a divisor, or whether some error terms are multiplied by 1/c. Of course, these are mathematically equivalent, but think being consistent in their presentation (e.g. showing c=16, or c=1/16 in both the text, figure and equations) would be more transparent for readers. I would also ask the authors to report the precise form of the cross-entropy error equation that they are using in the simulations including the "c" parameter and to consider how this relates to forms of the Cross-Entropy Error term used for multi-class and multi-label classification.

- I would also ask the authors to reflect on why this modified cross-entropy term was not necessary in order for the earlier EarShot model in the Magnuson 2020 paper to show appropriate cohort competition effects. I couldn't find a specific description of the error term used in this earlier paper, but I assume that this is cross-entropy error given the sparse binary output vectors used. Indeed, I know from personal experience that other (highly simplified) recurrent network models explored in the 1990s (definitely Davis et al, 1997, but perhaps also Norris, 1990; or Content & Sternon, 1994) used cross-entropy error with localist output units without this modification and still showed cohort competition effects (see Figure 2 and 3 of Davis et al, 1997 for example).

- It's not clear whether their model with appropriately tuned values of the "c" parameter (Figure 3I) still achieves probabilistic activation of target and competitor words (as explained in Davis et al, 1997, at each time point during a word, localist output units approximate the conditional probability of each word being present given the current input). I note that in Figure 3I the summed activation of target and competitor words clearly exceeds 1. I therefore think these output values no longer approximate conditional probabilities (which should, by definition sum to 1). Does this matter? Is it still the case that the other desirable properties of probabilistic activation (discussed in Norris & McQueen, 2008) still apply for this model? For example, will the model still show preferential activation of higher frequency words for cohort competitors that are mismatched in frequency (see (2) above)?

- I also wonder whether a training protocol that changes the way in which error terms are computed at specific points in a word could plausibly be implemented in the brain? Does this require that listeners anticipate when each word ends in advance? Or is there some other training protocol - e.g. back-propagation of error being applied only at word offset - that would help here? I note that there's a red box drawn at the end of each word's "training signal" in panel C of Figure 1 which could suggest this. A more detailed explanation of the time-course of the training procedure, and exploring whether model outcomes are sensitive to whether the pre-offset parameter is 200ms (rather than, say, 100 or 300ms), might be helpful in clarifying what's going on here.

(B) Relation to neural data

The most novel and important contribution of the present paper to the literature is that the authors use several different variants of their model - varying output representations, and model depth while keeping the total number of parameters relatively constant. They then explore whether measures of aggregate neural activity and change in neural activity in these different models predicts MEG data collected when participants recognise the same spoken words presented to the model. The approach used builds on previously published MEG data which was analysed with TRF-encoding methods by Gaston and colleagues (2023). They show that for (essentially) all of the simulations reported their neural network derived predictors substantially improve the amount of variance explained in MEG data compared to an acoustic "baseline" model which includes a spectrogram, onset-spectrograms, and word onset events. This success is impressive and builds on previously published demonstrations that measures of the internal operation of neural network models can help explain fMRI observations (Kell et al, 2018, Neuron). I had a few questions about the reason(s) for the success of their model in predicting neural data. I think some consideration of this issue would help ensure that the present work can be readily interpreted by readers who do not have access to the advanced simulation methods used in the present paper.

(4) What is the relative contribution of summed activity and change in activity? Can the authors report the variance explained by each of these different measures? I'm personally very interested in the idea that change in activity - perhaps related to moment-by-moment prediction error or surprisal - is especially predictive of neural responses. By separating the explanatory value of these two measures they might offer relevant evidence to assess predictive processing theories. I would also ask if summed and change in activity relate in the same way to neural data - for instance do both these model outcomes predict a positive impact on MEG responses? Or is a one a positive and the other a negative predictor?

(5) Another impressive aspect of the present work is their exploration of the impact of changes to network architecture on the neural variance explained. They persuasively show that deeper networks provide additional explanation of observed neural data, particularly when used with semantically structured, Glove output vectors. However, it's not sufficiently clear whether the improvement observed is due to deeper networks being more human-like, or whether this is due to their being additional parameters in the linear regression that relates network activity and neural data. For example, in Figure 2G is the marked improvement in variance explained due to separating two forms of hidden and output units? Or due to their being more free parameters (4 rather than 2) when using TRF regression to predict neural data? The same question could be asked when considering models at different depths (in Figure 3C)? Do their neural predictions from deeper models involve using separate predictors from each multiple layers? Or is it the case that it's only activity in the last layer of the deeper networks which is used to predict neural data?

(6) I will also admit that I'm unfamiliar with the separation - introduced in Figure 2E and other simulations - of local, recurrently connected hidden units, and other units that are connected to output units. Is this a common manipulation introduced in LSTM networks? How does this differ - in computational complexity and efficacy - from simply adding a second layer of hidden units? Is this difference between locally-connected and output-connected hidden units preserved in their later simulations? The presence of two graphs for "6-local" and "6-out" in Figure 4A suggests that this is the case. It would be helpful - in Figure 3A, for example - to show more details of the way in which all the deeper networks are organised and to discuss the difference between splitting a single set of hidden units, and adding an additional hidden unit later in more detail.

(7) In understanding why it is that later layers of deeper networks explain more variance in neural data it would be helpful if the authors could depict and explain the general properties of the aggregate activity and change in activity observed in the model during the timecourse of a spoken word. As depicted in Figure 4A, it seems that early layers that show a time-course correlated with amplitude envelope - is it possible to quantify this more directly? For later layers in these deep networks, units might show a less acoustically-responsive profile, but perhaps they could confirm this statistically (using correlational or RSA-type analyses).

Even with this explanation, however, it's still not clear why and how the temporal profile of later/deeper processing layers relate to position within a word. Does activity increase over time within each word? Or does activity relate to information theoretic or psycholinguistic predictor variables - e.g. word length, uniqueness point, cohort entropy, phonemic surprisal, etc. If it were possible to explain the operation of the neural network in these terms it would help open up this black box and to explain why these recurrent networks are such good linking hypotheses in explaining neural data.

(8) This challenge of relating different neural layers to specific neural sources is most clearly expressed in Figure 4D in which the authors explore whether different network layers predict activity in different neural regions. While the authors suggest that lower network layers predictive activity close to auditory cortex and higher network layers predict activity in other areas it seems just as accurate to say that the difference is more in the strength of correlation with neural activity as in the spatial location of predicted activity. In order to interpret these results as showing hierarchical correspondence, the authors need to confirm that there's a cross-over interaction such that early auditory areas are better explained by lower layers of the neural network and that higher lexical processing regions are better explained by higher layers of the neural network. The presentation of thresholded statistical maps can be persuasive to readers but does not suffice to confirm regional specialisation. What is required is the sort of analysis shown (for fMRI data, and using more conventional statistical modelling methods) in Figure 7 of de Heer et al (2017, Journal of Neuroscience). That is, confirming differential spatial tuning of relevant neural responses. In the absence of these analyses, a more cautious presentation is required.

Other/minor comments:

- Page 3-4 - the authors are right to talk about how many "transformer" models (e.g. BERT) make implausible assumptions about the nature of the temporal processing that is applied to speech or language stimuli (since left and right context must operate differently). However, they need to take care in distinguishing between different models since many current language models do not work backwards in time. My understanding is that all "generative" models necessary operate using "causal" mechanisms (ie. only forward in time). I've seen a distinction made between causal and acausal models and they might find these terms helpful here.

- Page 4, para 4 - they describe their model input as a "continuous spectrogram", but I think they're using discrete words (without coarticulation), and present single spectrogram slices at a time. It's not clear that "continuous" is a useful term here.

- Figure 2A / B / C - I found the depiction of these networks confusing. There's a level of "input" units that are shown as a graph. But, then there's a "dense" layer *before* the output graph. Does that mean the network has two layers of hidden units? I think not, but can't be sure. Furthermore, the labelling of the output as "dense" contrasts with the description of "Random Sparse Vectors" (RSV). EarShot similarly labels these outputs as "sparse". Is one of these labels in error?

- Figure 2E - dividing the recurrent layer into Hidden only and Hidden Unit + Output is seemingly not the same as having two HU layers in Figure 3. But why and how is the connection patterns and number of parameters different from the later simulations with two layers of hidden units?

- Figure 3 - when showing networks with "multiple hidden unit layers" (Figure 3A) I think it would be helpful to depict these in the same way as the depictions in Figure 2. For their best performing "3 layer" model in Figure 3D is this a model with "3 layers" (input / HU / output) or "3 HU layers" (input / HU1 / HU2 / HU3 / output)?

- Figure 3D / F / I - In plotting the output of these model I think that they're showing output activation for single, localist units. Is tht correct? Or is this a cosine similarity - like shown in similar Earshot simulations?

- Figure 4B/C - it's helpful to show similarity between different layers. It's striking to me that there's not always maximum similarity along the diagonal (that is, adjacent levels are not always maximally similar). Does this imply that there's processes that skip intermediate levels? Or does it suggest that (as argued by Kornblith et al), that simple correlations between layers is not necessarily a good measure of representational similarity? Also, it would be helpful to show similarity to input and output representations in these figures.

- Figure 4D - This figure aims to set out correspondance between early layers of a network, and the the spatial organisation of the auditory/speech/language processing hierarchy. I would be interested to know whether earlier and later TRF effects (ie. the temporal organisation of neural responses) similarly corresponds between their neural networks and brain responses.

Signed - Matt Davis, Cambridge, uK

**Have the authors made all data and (if applicable) computational code underlying the findings in their manuscript fully available?**

Reviewer #1: Yes

Reviewer #2: **No: ** They say that they will share model code on publication - and have a good track-record of doing this for previous, similar work. I therefore trust them to do this

PLOS authors have the option to publish the peer review history of their article (what does this mean? ). If published, this will include your full peer review and any attached files.

**Do you want your identity to be public for this peer review?** For information about this choice, including consent withdrawal, please see our Privacy Policy .

Reviewer #1: No

Reviewer #2: **Yes: ** Matthew H. Davis
---

## [Decision Letter · Decision Letter 1]

PCOMPBIOL-D-24-00643R1

Recurrent neural networks as neuro-computational models of human speech recognition

PLOS Computational Biology

Dear Dr. Brodbeck,

Thank you for submitting your manuscript to PLOS Computational Biology. After careful consideration, we feel that it has merit but does not fully meet PLOS Computational Biology's publication criteria as it currently stands. Therefore, we invite you to submit a revised version of the manuscript that addresses the points raised during the review process.

Please submit your revised manuscript within 30 days Apr 30 2025 11:59PM. If you will need more time than this to complete your revisions, please reply to this message or contact the journal office at ploscompbiol@plos.org. Please include the following items when submitting your revised manuscript:

We look forward to receiving your revised manuscript.

Kind regards,

Frédéric E. Theunissen

Academic Editor

PLOS Computational Biology

Thomas Serre

Section Editor

PLOS Computational Biology

**Additional Editor Comments:**

Dear Authors,

Your reviews were clearly well received. Could you please address the comment of Prof Matt Davis (Reviewer 2). I will read your answer carefully.

Thanks!

Frederic Theunissen

**Journal Requirements:**

1) Please upload all main figures as separate Figure files in .tif or .eps format. For more information about how to convert and format your figure files please see our guidelines:

2) Please ensure that the funders and grant numbers match between the Financial Disclosure field and the Funding Information tab in your submission form. Note that the funders must be provided in the same order in both places as well. State the initials, alongside each funding source, of each author to receive each grant. For example: "This work was supported by the National Institutes of Health (####### to AM; ###### to CJ) and the National Science Foundation (###### to AM).".

**Reviewers' comments:**

Reviewer's Responses to Questions

**Comments to the Authors:**

Reviewer #1: I appreciate the authors' efforts in revising the manuscript and commend them for addressing most of my previous concerns. I have only one more suggestion for further improvement:

The authors claim a progression of neural similarity with increasing layer depth. Is there a direct evaluation of the embedding space of the RNN layers? While Fig. 3B illustrates the WER across layers, which partially addresses this, it would be insightful to examine whether deeper layers capture more phoneme- or word-level structures compared to acoustic-level representations. To strengthen the analysis, the authors could consider performing the analysis in Fig 2F/2G across different depth layers. Or the authors may also consider a more direct evaluation similar to the approaches proposed by Pasad et al. (ASRU 2021) or Belinkov & Glass (NeurIPS 2017).

References:

Pasad, A., Chou, J. C., & Livescu, K. (2021). Layer-wise analysis of a self-supervised speech representation model. 2021 IEEE Automatic Speech Recognition and Understanding Workshop (ASRU), pp. 914-921. IEEE.

Belinkov, Y., & Glass, J. (2017). Analyzing hidden representations in end-to-end automatic speech recognition systems. Advances in Neural Information Processing Systems, 30.

Reviewer #2: Thanks for the opportunity to review this revised manuscript. As before, I am interested by the work and mostly satisfied with the revised manuscript that the authors have produced, and by their responses to my comments and suggestions.

I had only one substantive point in the section reporting "a modified loss for function for human like lexical competition" which I think the authors should do a bit more work to address. I also had a very few minor corrections and typos.

Presentation of lexical competition findings:

One strength of this manuscript is the way in which the authors take seriously the functional requirement that words be identified rapidly and in time with the speech input. This was not a feature of previous models (e.g. Kell et al, 2018, Neuron) which were otherwise modelled on human performance. It is very interesting to me that - by default - the model does not seem to simulate the appropriate time course for human speaker tokens, and only approximates this time-course for unseen synthetic speaker tokens. The correct, human-like timecourse becomes more apparent when the network reward function is modified to reduce the functional pressure against selecting a single word, and permitting parallel activation of multiple word candidates, this reduces word error rate and improves neural prediction.

However, I still feel that I don't understand a few details of the time-course of the activation profile for the original simulations, and for those with the modified reward function shown in Figure 4. Specifically:

- how is the original model able to identify target words so early - well before their phonemic uniqueness point? It is shown in the paper that target activation is delayed when human speaker tokens were untrained (Figure 4D). But, this still seems like very early activation compared to the more appropriate time-courses shown in Figure 4F (e.g. with C=256, or C=4096). I am troubled that it seems as if the model can predict which word is being heard before any disambiguating information has been heard! I would like the authors to keep this concern in mind in presenting their results. Are there additional details of the model, the speech tokens, or the timecourse of activity that are relevant here? If - as the authors state - this effect is not due to "overleaning of specific tokens" then what could be the cause of this?

- The presentation of the results in terms of activation values and probabilities is useful and potentially helpful. However, I still don't understand what is shown in Figure 4B, and similarly in the right hand plots of 4C, 4D, 4E, 4F). The values shown - ranging from 0 to 1 with multiple high values in the graph - do not correspond to what is described (values are individual word activation values divided by total activation and hence interpretable as probabilities which should sum to 1). I'm also confused by the fact that the target word is not included in these plots. If so, this is further reason to think that these can't be described as probabilities - the target word should be quite clearly the most probable candidate by word offset. Some changes or clarification would be very helpful.

- Furthermore, I see that the x-axis on the time-course plots in Figure 4 are given in milliseconds - this is useful since it (presumably) permits alignment with Figure 2E which similarly shows the time-course of speech presentation in milliseconds. I wonder if more could be done, though, to depict the nature of the target word and/or competitors - e.g. marking uniqueness points and phoneme onsets, or aligning identification timecourses to unique points. I do think it would be helpful for readers to more fully understand the difference between an appropriate and inappropriate time-courss of activation for simulations with the conventional and modified loss functions.

- In Figure 4G and 4H the black line shows word error rate and neural prediction for models without the additional divisor. This is labelled as "constant", but is perhaps more accurately labelled as c=1 or or using the "original" loss function.

- Finally, statements in the results are made with regard to improved word recognition and neural prediction with the modified loss function. However, reporting of the statistics is rather sparse. I wonder if there's some way of annotating, or marking those datapoints in Figure 4G and 4H that show significantly improved performance compared to the original loss function.

Minor comments:

- In Figure 2C and the main text there's an inconsistency in whether "Sparse Random Vectors" are abbreviated SRV or RSV.

- Figure 3D caption "because the predictive power"

- Page 10 line 4-5 "the pattern in response to synthetic speakers..."

- They have correctly described reference [48] as showing influences of subtle acoustic cues to word length. This conference paper is largely superseded by a longer journal paper (https://doi.org/10.1037/0096-1523.28.1.218) which reports these and other related results and might be more accessible for readers.

- In my original review, however, I referred to a different 1997 computational modelling paper (showing activation proportional to conditional probabilities in a localist RNN trained to recognise words in speech-like sequences):

Davis, M.H., Gaskell, M. G. and Marslen-Wilson, W. D. (1997) Recognising embedded words in connected speech: Context and competition in Bullinaria, Glasspool & Houghton (Eds) Proceedings 4th Neural Computation & Psychology Workshop: Connectionist Representations. London: Springer-Verlag. https://link.springer.com/chapter/10.1007/978-1-4471-1546-5_20.

I'm happy to share a PDF if this is unavailable to the authors. They're not obliged to cite this other paper. However, they might find it informative to consider the activation time-courses shown in this paper - and their clear link with activations corresponding with conditional probability - when thinking about my comments on Figure 4 above.

Signed: Matt Davis, Cambridge, UK

**Have the authors made all data and (if applicable) computational code underlying the findings in their manuscript fully available?**

Reviewer #1: Yes

Reviewer #2: **No: ** They say that they will share model code on publication - and they have a very good track-record of doing this for previous, similar work. I entirely trust them to do this

PLOS authors have the option to publish the peer review history of their article (what does this mean? ). If published, this will include your full peer review and any attached files.

**Do you want your identity to be public for this peer review?** For information about this choice, including consent withdrawal, please see our Privacy Policy .

Reviewer #1: No

Reviewer #2: **Yes: ** Dr Matthew H. Davis, MRC Cognition and Brain Sciences Unit, University of Cambridge, UK

**Figure resubmission:**
---

## [Editor Report · Decision Letter 2]

Dear Dr. Brodbeck,

We are pleased to inform you that your manuscript 'Recurrent neural networks as neuro-computational models of human speech recognition' has been provisionally accepted for publication in PLOS Computational Biology.

Best regards,

Frédéric E. Theunissen

Academic Editor

PLOS Computational Biology

Thomas Serre

Section Editor

PLOS Computational Biology

Thank you for diligently addressing all of the reviews. I am also sorry for the delay! I have gone over your article a couple of weeks ago and was convinced I had clicked on submit...

I have two minor comments:

1. P5. L 22.You says: “note that it is not possible to test the model on completely novel words because the mapping from speech to semantics is mainly arbitrary” I am not sure I understand exactly what you mean by “mainly arbitrary” and to what you are referring here exactly. Doesn’t GloVe, for example, encode semantic relationships (as you mention a few lines below p. 5 L 27-28) ? I know it is not trained on semantics but co-occurrence yields semantic relationships. Though these can be complex they are not what I would call arbitrary.

2. In your discussion you do mention the potential limits of your modeling in terms of not capturing linguistic feedback processing - which might be particularly important, as you mentioned, in more noisy environments. I would add that some of your conclusions will also depend on the data set. So the feed-forward (with local feedback) models will also be affected by noise, increase variability in natural conversational speech, etc. I would be surprised if the "localist" model remains the best performer in those situations.

Frederic Theunissen.

---

## [Editor Report · Acceptance letter]

PCOMPBIOL-D-24-00643R2

Recurrent neural networks as neuro-computational models of human speech recognition

Dear Dr Brodbeck,

I am pleased to inform you that your manuscript has been formally accepted for publication in PLOS Computational Biology. Your manuscript is now with our production department and you will be notified of the publication date in due course.

With kind regards,

Anita Estes
